# Hypothalamic Neuromodulation and Control of the Dermal Surface Temperature of Livestock during Hyperthermia

**DOI:** 10.3390/ani14121745

**Published:** 2024-06-09

**Authors:** Juliana Sarubbi, Julio Martínez-Burnes, Marcelo Daniel Ghezzi, Adriana Olmos-Hernandez, Pamela Anahí Lendez, María Carolina Ceriani, Ismael Hernández-Avalos

**Affiliations:** 1Department of Animal Science, Federal University of Santa Maria, Av. Independência, Palmeira das Missões 3751, RS, Brazil; 2Facultad de Medicina Veterinaria y Zootecnia, Universidad Autónoma de Tamaulipas, Victoria City 87000, Mexico; 3Animal Welfare Area, Faculty of Veterinary Sciences (FCV), Universidad Nacional del Centro de la Provincia de Buenos Aires (UNCPBA), University Campus, Tandil 7000, Argentina; ghezzi@vet.unicen.edu.ar; 4Division of Biotechnology—Bioterio and Experimental Surgery, Instituto Nacional de Rehabilitación-Luis Guillermo Ibarra Ibarra (INR-LGII), Mexico City 14389, Mexico; 5Faculty of Veterinary Sciences (FCV), Universidad Nacional del Centro de la Provincia de Buenos Aires, CIVETAN, UNCPBA-CICPBA-CONICET (UNCPBA), University Campus, Tandil 7000, Argentina; 6Facultad de Estudios Superiores Cuautitlán (FESC), Universidad Nacional Autónoma de Mexico (UNAM), Cuautitlán 54714, Mexico

**Keywords:** heat stress, livestock, heat defense behavior

## Abstract

**Simple Summary:**

This review aims to discuss the hypothalamic control of hyperthermia in livestock, including the main physiological and behavioral changes that animals adopt to maintain their thermal stability. This review analyzes the main changes that can be observed in livestock when exposed to intense heat. Vasodilation, sweating, and behavioral modifications (e.g., standing posture, shade seeking, feed intake, among others) are the results of neuronal projections from peripheral thermoreceptors to the preoptic area of the hypothalamus and effector neurons. Understanding these mechanisms can help to visualize their effect to restore thermal neutrality in livestock.

**Abstract:**

Hyperthermia elicits several physiological and behavioral responses in livestock to restore thermal neutrality. Among these responses, vasodilation and sweating help to reduce core body temperature by increasing heat dissipation by radiation and evaporation. Thermoregulatory behaviors such as increasing standing time, reducing feed intake, shade-seeking, and limiting locomotor activity also increase heat loss. These mechanisms are elicited by the connection between peripheral thermoreceptors and cerebral centers, such as the preoptic area of the hypothalamus. Considering the importance of this thermoregulatory pathway, this review aims to discuss the hypothalamic control of hyperthermia in livestock, including the main physiological and behavioral changes that animals adopt to maintain their thermal stability.

## 1. Introduction

Exposure to high ambient temperatures can be implicit inside production systems due to the design of the units or the geographical location of farms, making livestock susceptible to heat stress and hyperthermia [1,2]. Moreover, climate change and global warming are current challenges for farm animals [3,4]. Preventing increases in core body temperature is essential because several physiological and behavioral mechanisms are activated when animals are under thermal stress [5]. When not modulated, these mechanisms can negatively affect the health and productive performance of livestock. Thus, understanding these responses is essential to promote thermal stability in animals.

Maintaining a regular body temperature depends on the interaction between internal and external factors, which is also influenced by the size, weight, morphology, physiological state (e.g., gestation, estrous cycle), and the metabolic rate of animals [6,7,8]. Generally, when mammals experience an increase in core body temperature, two main mechanisms are activated: physiological and behavioral. In the first, physiological responses include involuntary autonomic adaptations such as sweating, respiratory evaporation, and changes in blood perfusion to facilitate thermal exchange (e.g., vasodilation) [9,10,11]. On the other hand, behavioral modifications include compensatory changes in body posture (e.g., increasing their standing time or lying on the side) or shade-seeking and increasing water intake to promote heat loss and limit heat production [10,12,13].

Regardless of the physiological and behavioral adaptation, in mammals, the main thermoregulatory center is the hypothalamus, particularly the preoptic area (POA) of the hypothalamus, where peripheral thermal afferents and thermoregulatory efferents are processed [5,14,15]. This review aims to discuss the hypothalamic control of hyperthermia in livestock, including the main physiological and behavioral changes that animals adopt to maintain their thermal stability. 

## 2. Search Methodology

A literature review was conducted using Web of Science, Scopus, Science Direct, and PubMed. The keywords used for the search were “hyperthermia”, “heat stress”, “transport stress”, and “infrared thermography”. The selected articles were studies or review papers published from 2000 to 2024 on animals with heat stress where IRT was applied as a complementary monitoring tool.

## 3. Hypothalamic Control of Hyperthermia: Peripheral Perception of Thermal Stimuli and Supraspinal Processing

### 3.1. Peripheral Thermal Receptors

Specialized receptors that transduce cold and heat stimuli into electrical signals (called thermoreceptors) detect changes in the external and internal temperature of the organism [16,17]. When animals are outside their thermo-neutral zone (a temperature range where mammals can maintain a balance between heat production and dissipation) and exposed to temperatures that exceed their maximum temperature, several thermoregulatory mechanisms are triggered to increase the amount of heat dissipated from the body or to decrease the amount of heat produced by the animals [18].

Both physiological and behavioral changes are directly related to the activation of thermoreceptors located mainly in the skin, where free nerve endings detect thermal variations [19,20]. Lezama-García et al. [21] describe that thermosensory neurons respond to both hot and cold stimuli, activating ion channels so the neurons can transmit the information to spinal and supraspinal structures [22]. In particular, the main thermoreceptors comprise the transient receptor potential (TRP). TRPs are cation channels formed by transmembrane proteins that act as transducers through changes in membrane potential due to the intracellular Ca^2+^ concentration. Most react to fluctuations in temperature, pH, and osmolarity; however, injury, pain, pheromones, flavors, taste, ionic imbalances (Ca^2+^), volatile compounds, mechanosensations, and cytokines can also trigger them. Since their discovery, about 30 trp genes and over 100 TRP channels have been identified and categorized into their respective families and subfamilies [21].

Six subfamilies are identified as thermosensors (TRPV1-4, TRPM2-5,8, TRPA1, and TRPC5). Neuronal afferents that perceive heat and cold can be differentiated by presenting specific TRP channels for each stimulus because their activation depends on specific thermal properties [23]. Most neurons that allow the identification of differences in the range of −10 to 60 °C are thermosensitive receptors located in nerve endings that are found in the trigeminal ganglion that innervates the face and head, as well as in the ganglia of the dorsal horn of the spinal cord [24].

Based on the properties of TRP thermosensors, four basic thermal sensations are recognized: (1) cold, −10 to 15 °C; (2) cool, between 16 and 30 °C; (3) warm, 31 to 42 °C; and (4) hot, 43–60 °C. Depending on the stimulus, PRTs are classified into hot- and cold-sensitive receptors [21,25]. Regarding the heat-sensitive receptors, TRP vanilloid V1 and V1 perceive harmful heat above 43 °C and 52 °C, respectively [23]. Moreover, TRPV3 can also perceive harmless heat with warm temperatures above 34 °C (ranging between 23 and 39 °C) [26], while TRPV4 is activated with moderately warm temperatures of 27 to 34 °C [23]. Molecular studies have shown that TRPM8 participates in cold-defensive responses, while TRPM2, TRPM3, TRPM4, and TRPM5 are activated by heat. Particularly, TRPM3 responds to harmful heat. One difference between TRPM4 and TRPM5 and the others is that they are not permeable to Ca2+, though they participate in membrane depolarization [27,28].

Current research regarding warm-sensitive thermoreceptors is still ongoing. However, it has been suggested that TRPM2 may be involved in the activation of thermoregulatory behavioral responses in warm environments [29]. Nozadze et al. [30] suggest that the co-expression of TRPV1 and TRPA1 have a synergistic relationship to process thermal stimuli. This was evaluated in adult male Wistar rats, in whom the effect of mustard oil and capsaicin increased heat hyperalgesia at 30 °C. In another study performed by Cheng et al. [31], it was identified that heteromeric TRPV1 and TRPV3 channels actively participate in the detection of these stimuli. The authors observed that the activation threshold for TRPV1 channels is at 37.7 ± 0.7 °C, while TRPV3 channels activate at 31.3 ± 0.6 °C. This suggests that the co-expression of these channels contributes to heat perception, as also described by Lei et al. [32]: when identified in cell cultures of 4–5 week-old mice, the expression of a transmembrane protein called TMEM79 acted as a negative regulator of TRPV3. They also observed that in cells lacking TMEM79, the flow of Ca^2+^ ions was enhanced and was a process mediated by TRPV3. They also observed that mice with null expression of TMEM79 preferred higher temperatures than mice that presented this protein due to elevated TRPV3 function. This possibly supports the role of this molecule in the transduction of hot thermal signals and even possibly suggests that TRPV1/TRPV3 are unique heat sensors that contribute to the adjustment of sensitivity to the sensation of heat.

These channels transduce thermal signals from the skin to subsequent central regions such as the dorsal root ganglion to thermoregulate. In the spinal cord, it has been described that TRPV1, TRPV3, and TRPA1 channels are also expressed [33,34]. The activation of these ion channels is the first step to trigger the compensatory responses to dissipate heat by its connections to the main thermoregulatory center, the POA.

### 3.2. Hypothalamic Modulation of Hyperthermia

Physiological and behavioral changes depend on the activation of peripheral thermoreceptors, hypothalamic integration, and subsequent connections to the mesencephalon and spinal cord [6]. Thermoreceptors transduce the information and synapse with spinal dorsal horn neurons, where specialized warm-sensitive and cold-sensitive neurons located in the lateral parabrachial nucleus (LPB) respond to thermal stimuli [28,35]. The neurons located in the dorsal part of the LPB (LPBd) are activated by warm inputs (~36 °C) to elicit autonomic heat-defense responses [5,35]. 

LPBd neurons are glutamatergic, and their inactivation causes impaired autonomous thermoregulatory responses to avoid heat or cold stimuli [35,36]. Through optogenetics, it has been observed that LPB glutamatergic neurons express dynorphins and cholecystokinin to defend the organism from hyperthermia [37]. In rats, these neurons inhibit thermogenic mechanisms such as the activation of the brown adipose tissue while promoting tail vasodilation [37,38]. 

To modulate these responses, the LPBd projects to the POA of the hypothalamus. This area is the primary integrative center of thermoregulation, where 30% of the thermosensitive neurons are located [14,15]. These neurons are GABAergic and have glutamatergic and prostaglandin (PG) receptors [5], which are stimulated by changes in hypothalamic and central temperature and the temperature recorded in other anatomical regions such as the spinal cord, blood, and viscera [39,40,41]. In this sense, the role of other substances such as PG, nitric oxide, histamine, and tumor necrosis factor needs to be highlighted since, for example, PGE2 reduces the activity of GABAergic neurons in the hypothalamus and can elicit vasoconstriction and thermogenesis in rats [42]. In the same species, Nakamura et al. [43] found PG EP3 receptors in the POA. When stimulating PG EP3-expressing neurons, the body temperature decreases, while hyperthermia can be triggered by inhibition of the same receptors, showing that GABAergic signaling in the POA is fundamental to maintaining thermal homeostasis. Furthermore, the release of nitric oxide from endothelial cells is a potent vasodilator that can contribute to heat loss in animals suffering from hyperthermia [44].

Thermal information from these structures is transmitted by the vagus nerve (X cranial nerve) to the POA through the nucleus of the solitary tract in the caudal portion of the medulla oblongata for projection to the hypothalamus [36,45]. The spinal-LPB-POA circuit is considered the main thermosensory pathway to modulate the adaptative mechanisms to environmental thermal challenges [35,46,47]. In mouse models, opioidergic connections are present between the LPB to the POA, and studies have revealed that warm stimuli activate neurons expressing dynorphin and enkephalin, which participate in heat defense behaviors [36,48,49]. Nonetheless, other regions also participate in thermal perception, such as the afferent pathways from the LPB to the dorsomedial hypothalamus (DMH) and the rostral medullary raphe region (rRM) that includes the rostral raphe pallidus nucleus (rRPa), the adjacent raphe magnus nucleus (RMg), and the lateral pyramidal nucleus [14,50]. 

The LPB projects to the median preoptic nucleus (MnPO), mediating ascending heat-related responses [50]. The GABAergic projection neurons of the POA connect two nuclei: the MnPO and the medial preoptic area (MPO). It has been observed that neurons in these two regions are sensitive to heat, although MPO activity is greater at thermoneutral temperatures [51]. Furthermore, connections from the LPB to the lateral hypothalamus (LH) are required to modulate thermal stimuli [46]. There are also connections from the LPB to the periventricular nucleus (PVN) through the DMH [36]. The DMH projects glutamatergic neurons to the rRM and its different nuclei, which contain sympathetic premotor neurons that project to the sympathetic preganglionic neurons of the intermediate-lateral column of lamina VII of the spinal cord (IML). The activity of IML neurons depends on the secretion of glutamate and serotonin (5-HT) in the rRM. However, 5-HT modulates the excitatory activity of glutamate [14,52]. 

The projections of the POA connect with the neurons of the DMH and to the rRM without depending on the activity of the DMH, so it is thought that these direct or indirect connections of the POA-rRM trigger different responses. For example, the direct synapse of POA neurons to those of the rRM results in cutaneous vasoconstriction as a febrile response against cold, while the POA-DMH-rRM connection produces tachycardia and thermogenesis [4]. Both adrenergic and cholinergic fibers modulate blood flow, producing vasoconstriction and vasodilation, respectively [44]. Therefore, sympathetic cholinergic nerves produce cutaneous active vasodilation to dissipate heat [53]. Some reports have shown that transdermal application of acetylcholine and local heat (41 °C) cause hyperemia of the microvasculature of the skin [54]. Moreover, these nuclei are involved in the cutaneous vasomotor response and sweating response, controlling skin blood flow and BAT metabolism. These aspects will be addressed in the following sections (Figure 1) [55,56,57].

## 4. Response to Hyperthermia: Systemic and Local Changes

In various circumstances, it has been documented that an increase in the body temperature of animals is associated with warm environmental conditions with high humidity and solar radiation [58]. Livestock shows a greater susceptibility to thermal stress when exposed to these climatic conditions and when maintained in intensive production systems where growth rates are sought to be maximized. Particularly in livestock, the consequences of hyperthermia include health problems, productivity, and deaths during extreme events, as well as serious economic losses [59].

Several POA neurons are known to modulate heat defense responses, including vasodilation and behavioral modifications such as reduction of physical activity and energy expenditure, body posture changes, and suppression of thermogenesis both by brown adipose tissue or shivering [37]. Cattle exposed to heat stress can also alter their endocrine and immune responses [60]. These mechanisms prevent further increases in core body temperature to restore homeostasis.

### 4.1. Vasomotor Response to Hyperthermia: Vasodilation

The immediate physiological response of animals to thermal stress involves altering the diameter of superficial blood vessels, facilitating heat exchange in skin tissue with the environment through radiation [49,61]. This effect is facilitated in regions with glabrous skin and dense anastomosis (regions called thermal windows that are schematized in Figure 2) [5].

An example of this can be observed in Rodríguez-González et al.’s [62] study, where the effect of 2 h transport in the surface temperature of buffaloes was evaluated. In general, increases in the surface temperature above 4 °C were recorded for the lacrimal caruncle, periocular, lower eyelid, auditory canal, nostrils, and body regions such as the abdominal, lumbar, and parietal region after transport. A similar study made by the same researchers compared the effect of short (50.33 min.) and long journeys (13.31 h) on the thermal response of 1516 water buffaloes. They observed that the surface temperature of the periocular, lacrimal caruncle, nasal, auricular, frontal–parietal regions of the head, pelvic limb, abdominal and thoracic regions, and lumbar region increased by 3 °C during short journeys, a response that was not observed during long ones [62]. 

The physiological explanation for the increases in surface temperature is due to the peripheral vasodilation that is present when animals are exposed to intense heat. Peripheral vasodilation enhances heat dissipation and, therefore, helps to reduce core temperature [63,64]. This vasomotor response also responds to acetylcholine release and other co-transmitters from sympathetic cholinergic nerves, increasing cutaneous blood flow up to 7700 mL/min [19,63]. Moreover, other mediators such as vasoactive intestinal peptide, substance P, histamine, prostaglandins, and TRPV1 activation participate in vasodilation [65,66]. 

Understanding this mechanism allows us to explain the differences found in body regions for heat elimination. In a study where short and long journeys were compared to evaluate the thermal response of water buffaloes, the authors also found that the surface temperature of central regions was 1 °C higher compared to regions such as abdominal or lumbar [62]. The thermal capacity of certain regions could be related to the difference in the arrangement of blood capillaries, results that have been reported when comparing the surface temperature between the eye region and peripheral regions, finding that central regions are at least 1.5 °C above the temperature at the limbs of dogs and large ruminants [67,68] or the eye region when compared to the tail temperature in rodents [69]. Figure 3 shows the comparison between central and peripheral thermal windows in water buffalo.

Another possible example of this is the parietal region in different species of ruminants, where it has been suggested that horns might participate in thermoregulation [70]. A study conducted by Algra et al. [71] evaluated the thermoregulatory function of the horns in 18 cows from 3 farms in a temperate climate area. They observed that the temperature in the horns increased by 0.18 °C for each unit of the heat load index. They also reported that dehorned animals had a significantly higher eye temperature than animals with horns (by 1 °C). These results support the idea of the participation of horns in thermoregulation in this species. However, they also show the importance of the arrangement of the blood vessels to thermoregulate. In this sense, the parietal region is irrigated by the superficial temporal artery (*temporalis superficialis*), *transversa faciei*, *auricularis rostralis*, *palpebralis inferior lateralis*, and *palpebralis superior lateralis*, blood vessels that provide circulation to the horns [72].

One of the tools used to evaluate dermal surface temperature as an indicator of well-being in livestock is infrared thermography (IRT). Thus, various studies have exposed the possible uses and applications of IRT—for example, in the phenomena of hyperthermia associated with stress in lactating dairy cows during isolation challenges [73], diagnostic support of subclinical mastitis not only in cows but also camels (*Camelus dromedarius*) [74,75], and febrile processes due to diseases and in the evaluation of negative emotions such as fear and anxiety [9,76]. All these cases were evaluated indirectly by the vasomotor response associated with the increase in surface temperature.

Regarding other ruminants, in sheep, this tool has been used to determine and evaluate hoof injuries, where a difference of 8.5 °C could be observed between healthy and diseased hooves, showing a sensitivity of 92% and a specificity of 91% in the diagnosis of this disease [77]. In this same species, IRT of the ocular surface has been useful as a non-invasive method to measure the level of thermal stress developed during shearing, resulting in a tool that was positively correlated with rectal temperature and serum cortisol levels [78].

Likewise, similar findings have been reported in other species, where the possibility of correlating the environment’s high temperatures with rabbits’ health can be highlighted by correlating physiological responses with fecal cortisol levels [79]. Another example of application is in horses, where IRT seems to be a good indicator of physiological stress, which is why it can be useful to evaluate the effectiveness of training in endurance horses by measuring the temperature of the ocular surface, crown, gluteal muscles, and *longissimus dorsi* since the eye temperature could be correlated with the heart rate (r = 0.42 left eye; r = 0.48 right eye), while the frontal crown temperature had a positive correlation with cortisol levels (*p* < 0.01) [80]. Thus, IRT can become a useful tool for the early identification of horses that are not suitable to compete or continue in competition.

On the other hand, during routine procedures such as teeth clipping, dehorning, or tail docking, local vasodilation can also be observed as a result of possible acute pain perception, as observed in 1-day-old crossbred piglets (White Large × Landrace) undergoing teeth clipping [81]. A progressive increase in the upper lip temperature (by 0.7 °C) was recorded, possibly associated with vasomotor changes. Contrarily, although no differences were found in the soft tissue of piglets subjected to teeth grinding, the tooth temperature during the procedure reached up to 90 °C [82]. In dehorned animals, regardless of the use of oral and local analgesics, the temperature on the dehorning wound showed a progressive increase from day one (38.83 ± 0.42 °C) to day seven (40.30 ± 0.40 °C), a response that might be associated with inflammatory substances (e.g., PG) and local vasodilation [83]. Regarding tail docking, an increase in the surface temperature of the ocular region of crossbred piglets (by 1.1 °C) docked with side-cutter pliers was observed in these animals when comparing the before and after periods [81]. A similar response was observed in daily cattle subjected to tail docking. The surface temperature of the tail of docked animals was 1.43 °C warmer than the values registered in non-docked cattle, possibly as a result of local vasodilation and hypersensitivity [84].

In summary, the general vasomotor response in domestic animals can be considered a primary and passive method that increases blood flow to promote heat loss. The capacity of the different body regions to dissipate heat seems to be closely related to the arrangement of blood vessels, which is why it is suggested that in some species, the use of horns may be a key element in thermoregulation during hyperthermia.

### 4.2. Sweating

Another thermoregulation mechanism present in livestock during hyperthermia is sweating to lose heat through evaporation [19]. Sweat is released by eccrine glands distributed on all body surfaces. Their activation is mediated by sympathetic cholinergic fibers [85]. Although it is a mechanism particularly used to increase heat loss, it is not present in all species due to the difference in the arrangement and development of the sweat glands. An example is the Egyptian water buffalo, in whom the structure and dimension of sweat glands were studied and compared to *Bos* cattle by Hafez et al. [86]. The number of glands per cm^2^ of skin was lower in buffalo (394 gland/cm^2^) than in cattle (2633 gland/cm^2^). This trait might be related to the thermal tolerance in the animals, as addressed by a study performed in 10 female Murrah buffaloes exposed to hot and humid environments. A negative correlation between respiratory frequency and skin thickness was observed (r = −0.73, *p* = 0.015). The authors conclude that water buffaloes resort to polypnea as a heat-related tolerance response that also needs to be enhanced by providing shade [7].

This study shows the importance of sweating as an essential mechanism for thermoregulation in animals during hyperthermia. However, the differences in distribution between species limit its effectiveness, and it is suggested that equidae, bovidae, and primates are the only species that have sweat glands [87]. In particular, for horses, sweating dissipates 70% of the heat loss during exercise due to evaporation. In this species, it has even been found that the sweat rate (L/h/m^2^) is three times greater in the horse compared to the human with a similar exercise intensity [17,88,89].

In this regard, there are not only distribution differences between species but also differences in their location site. A histological analysis of water buffalo’s sweat glands was performed by Debbarma et al. [90]. It was reported that the number of sweat glands/cm^2^ in the head, neck, and tail regions was 20–30% higher compared to the thorax and abdomen region. This difference might impact not only the effectiveness of thermoregulation strategies in animals but also make it evident that animals use specific anatomical regions to thermoregulate. Thus, Barros et al. [91] evaluated the surface temperature in ten Murrah buffalo bulls in four anatomical parts (ocular, right flank, left flank, and scrotum). Although they observed a strong positive correlation between temperature and the temperature humidity index (THI) in all regions (ocular = 0.72, right flank = 0.77, left flank = 0.75, and scrotum = 0.41), the highest temperature was in the ocular region. This could be explained by the difference in the distribution of the number of sweat glands between regions and even in the same distribution of superficial blood capillaries, which would facilitate the thermoregulation capacity in the animal.

Therefore, sweating is an essential method used to recover thermoneutrality in animals in species that have an abundant arrangement of sweat glands in the skin. For this same reason, the presence of these same glands helps us understand why some species are more susceptible to thermal stress or have less-effective thermoregulation systems compared to other species.

### 4.3. Endocrine and Immunological Response Associated with Hyperthermia

Endocrinological changes play an important role in the metabolic response to heat stress through the neurosecretion and modulation of hormones, especially glucocorticoids, antidiuretic hormone, growth hormone, thyroxine, prolactin, and aldosterone, since they represent an adaptive mechanism of the organism regarding this condition [92,93]. For example, in hyperthermia due to acute heat stress, the concentrations of antidiuretic hormone, prolactin, glucocorticoids, and catecholamines increase, while those of aldosterone decrease. However, during chronic heat stress, levels of cortisol, growth hormone, and thyroxine may remain unchanged or may even decrease [60,94]. Changes in hormone concentrations during acute heat stress events are associated with decreased electrolyte concentrations and fluid balance, as water and electrolyte losses occur during panting and sweating due to glucocorticoid secretion. In this regard, it has been reported that elevated prolactin concentrations play an important role in sodium and potassium balance at elevated ambient temperatures, given its relationship with aldosterone secretion [95,96]. Derived from the action of glucocorticoids, catecholamine concentrations have been reported to be elevated during chronic heat stress events, possibly due to increased activity of sweat glands regulated by sympathetic innervation [97].

Hyperthermia due to stress and climate change means that dairy cows may be more susceptible to diseases that are usually positively associated with the survival of microorganisms and/or their vectors. These diseases can increase heat stress-related immunosuppression, negatively affecting the immune system through cellular and humoral immune responses mediated by the hypothalamic–pituitary–adrenal axis (HPA) and the sympathetic–adrenal medulla axis (SAM) [98]. The main responses are associated with an increase in blood cortisol levels, which is associated with the greater production of cytokines such as interleukin-4 (IL-4), IL-5, IL-6, and IL-12 [99]. Similarly, stress hyperthermia has been shown to inhibit interferon-gamma (IFNγ) and tumor necrosis factor α (TNF-α). The effects of heat stress on the immune response of livestock can be reduced by improving current animal-selection methods, as well as the further development of breeds resilient to climate change, which can support the sustainability of future livestock systems of animal production [100].

## 5. Behavioral Changes Observed in Domestic Animals Experiencing Hyperthermia

Apart from the physiological modifications, animals adopt certain behavioral changes or body postures to face hyperthermia and increase heat dissipation via radiation, convection, or evaporation [101,102,103,104,105,106]. These behaviors can be classified into those that promote heat dissipation (e.g., shade-seeking, changes in body posture) or those that limit heat production (e.g., reducing feed intake and locomotor activity) [35,107,108,109,110].

Regarding shade-seeking, several studies have reported the benefits that shaded areas have on animals’ thermoregulation, reducing the heat load in cattle by about 45% [111,112]. Gaughan et al. [113] reported that providing shade reduced the rectal temperature of Angus heifers (between 38.8 ± 0.8 and 39.4 ± 0.8 °C) in comparison with heifers in unshaded pens (between 39.1 ± 0.8 and 39.7 ± 1.8 °C). Similarly, crossbred calves maintained under agro-net shading spaces had lower rectal temperatures (minimum of 38.99 °C) and respiratory rates (minimum of 20.83 rpm) than animals housed in places with roofs of other materials [114]. The same results were observed in sheep, animals who prefer shaded areas (56–43%), and this reduces the respiratory rate, a sign of thermal stability [115,116].

Changes in standing and lying position are also observed in animals facing intense heat to facilitate heat dissipation. Standing increases the exposure of skin to airflow, maximizing evaporation [104,110], and animals can increase this posture by up to 10% when exposed to a heat load above 15% [117,118]. So, cattle exposed to a THI > 72 (considered intense heat) and with core body temperatures above 38.8 °C increased the likelihood of cows standing by 50% [119]. Similarly, Herbut and Angrecka [120] reported in Holstein-Friesian dairy cows that a THI > 73 had a significant and inversely proportional correlation with lying posture, where cows decreased their lying time from 11.3 to 9.4 h/day. The same observation was discussed by Hut et al. [121] in dairy cows; however, the authors reported that cows start to show adaptative behaviors (increased time standing and reduced rumination time) from an average ambient temperature of 12 °C and a THI of 56, showing that animals do not need to reach critical temperatures to start thermoregulatory behaviors when their rectal temperature increases.

In Suffolk and Dorset sheep, the same reaction was observed in Reis et al.’s [122] study when comparing the behavioral responses of animals at ambient temperatures of 20 and 35 °C. The authors found that higher temperatures (35 °C) increased the total time of standing (355 vs. 486 min) and decreased eating (932 vs. 727 min). Likewise, the lying behavior of ewes and rams increased by up to 45.10 ± 2.42% during severe heat stress [123]. In the case of pigs, the animals increased their lying time on the side by up to 5.9% to promote a larger contact area of the skin surface with the floor to dissipate heat through conduction [124]. In these latter cases, the behavior may be contradictory; however, heat transfer by conduction and evaporation helps explain why the rest time was longer in these animals, as discussed by Gómez-Prado et al. [125], who describes that pigs have a very low density of sweat glands (30/cm^2^) compared to species such as cattle (800–2000/cm^2^), which causes the pig to lose little heat when sweating. Another factor to consider is subcutaneous fat; for example, pregnant or lactating sows and commercial-weight pigs require greater energy consumption during their productive stage, so this effect decreases the heat dissipation capacity.

The importance of coat/akin color has been previously reported by Arp et al. [126], who highlighted that the surface temperature of black-haired cattle can be 21 °C above the values registered in light-haired animals due to higher absorptivity. Another example is dark-skinned pigs (Creole breed), animals that spend 4% more time lying down than light-skinned pigs with lighter skin. In multiparous Holstein cows, Anzures-Olvera et al. [127] found that when comparing black- and white-colored coats, those with a black coat color had a higher RT (0.1 °C) and also had increased mean corpuscular volumes (dark cows: 54.7 fL vs. white cows: 53.8). Moreover, cows with a white coat produced 394 kg more fat-corrected milk.

Among the strategies to reduce heat production, reducing energy consumption and locomotor activity stand out. Another strategy to reduce heat production is limiting feed intake [128]. Ruminants reduce their grazing time and feed intake when they are outside their comfort zone (12–24 °C) [110]. This behavior has been noted in 90% of water buffaloes, which opt to graze at 7 am—seen as the cooler part of the day—and allocate around 76% of their time budget for resting during the hottest hours of the day [129]. In Holstein dairy cows under an ambient temperature of 28.9–30.5 °C, Spiers et al. [130] found significant correlations between increases in rectal temperature, milk yield, and feed intake. When dairy cows had an average rectal temperature of 39 °C, reduced dry matter intakes (DMI) by 14.6 kg/day and milk yields by 11.8 kg/day were recorded. Similarly, in mid-lactation cows, Chen et al. [131] determined that for each unit increase in THI, dry matter intake (kg/day) decreased from 21.3% in thermoneutral cows to 17.3% in heat-stressed animals, which also reduced milk yield (32.5 kg/day vs. 27.0 kg/day). Decreasing feeding intake is a strategy of mammals to reduce their metabolic rate and heat load, limiting heat production and preventing a further increase in core body temperature [132].

Contrarily to feed intake, animals significantly increase their water intake [133], as observed in Angus crossbred yearling steers, in whom water intake increased by 6.8 L/animal [134]. Moreover, another activity that involves water is wallowing. Wallowing is another behavior that is particularly adopted by water buffalo when exposed to heat stress [103]. Wallowing helps to decrease body temperature by increasing heat dissipation via evaporation, as shown in a study assessing the thoracoabdominal surface temperature of water buffaloes [135]. The authors observed that ambient temperatures of 38 °C increase the average surface temperature of buffaloes up to 36.4 °C, a value that can decrease up to 4 °C after wallowing. Water buffalo spend an average of 2.35–2.91 h a day performing this behavior [105,136], and some studies have found increases to 4.06 h when buffalo heifers are exposed to intense heat (>35 °C) in silvopastoral systems [105]. These examples help to understand how wallowing is a thermoregulatory behavior that enhances heat loss via evaporation and vasodilation and prevents further increases in body temperature [136]. Table 1 shows the main physiological and behavioral changes in animals experiencing hyperthermia.

### Management of Heat Stress in Livestock Production Systems

Stress-induced hyperthermia events in livestock are characteristic of a process of maladaptation to environmental factors or deficiencies in thermoregulation; therefore, the veterinarian’s attention should focus on prevention and risk reduction. Among the main suggested actions for the management of heat stress are the cessation of livestock movement, provision of supplemental water sources in pens, cooling of the pen surface, improvement and management of nutrition and feeding plans, removal of manure from pens, and the provision of shade, which have been demonstrated to be beneficial strategies for livestock exposed to hot climates or thermal stress events [134,137,138,139,140].

## 6. Perspectives

It has been discussed that animals undergoing hyperthermia trigger hypothalamic thermoregulatory mechanisms to coordinate vasomotor and sweating activity, as well as behavioral modifications. However, there are still remaining areas of research, such as the multi-modal participation of thermoreceptors TRPV1, TRPA1, and TRPV3 [31,141]; studying the activation threshold of said receptors to detect thermal stimuli and how this is different from when these receptors act as nociceptors could help to understand their role during both physiological conditions.

An interesting aspect to address could be the implementation of biomarkers such as amino acids and neuropeptides that can modulate the tolerance to heat stress in birds [142]. Regarding this, in ruminants, so-called heat shock proteins (HSF1, HSP70, and HSP90) have been proposed as a method to differentiate between hyper and fever, together with other elements such as glutathione peroxidases, superoxide dismutase, and malondialdehyde [143].

On the contrary, animal temperature-evaluation methods currently rely heavily on invasive techniques like rectal thermometers, which can induce acute stress during handling. Incorporating infrared thermography could aid in non-invasively detecting temperature rises [144,145,146]. Thus, it is crucial to validate this technique to recognize and diagnose states of hyperthermia or thermal stress, possibly allowing the development of an automatic tool with the considerable advantage of being non-invasive. In this sense, with this same tool, it is possible to identify a more appropriate anatomical region or thermal window for each species to adequately monitor their thermal state.

## 7. Conclusions

The main thermoregulatory responses in animals during hyperthermia are physiological and behavioral changes modulated by the thermal circuitry between peripheral thermoreceptors, the POA, and afferent fibers to effector organs. The connection between the POA and efferent pathways is essential for animals to thermoregulate. Among the physiological responses, vasodilation is a mechanism that increases heat dissipation to reduce heat load and decrease the core body temperature. Sweating also promotes heat loss by evaporation together with polypnea. Regarding behavioral changes, livestock increase their standing time to increase their skin surface’s exposure to airflow. Moreover, they engage in other behaviors to limit metabolic heat production, such as decreasing feed intake and locomotor activity. These responses aim to restore thermoneutrality and prevent further increases in the core body temperature. Understanding that livestock has a thermoneutral zone where these responses are not triggered is essential to preventing hyperthermia in farm animals.

## Figures and Tables

**Figure 1 animals-14-01745-f001:**
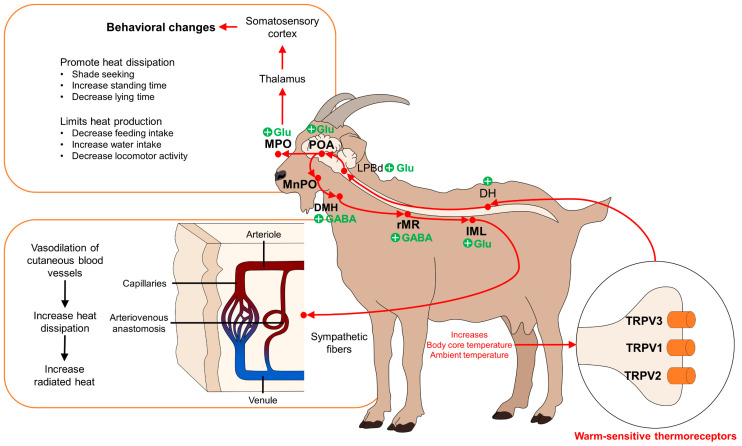
Hypothalamic modulation of temperature during hyperthermia. Warm-sensitive thermoreceptors detect increases in core body temperature or ambient temperature. These stimuli are transmitted to the DH of the spinal cord to subsequently project to the LPBd, the region that is particularly activated with heat-related signaling. From the LPBd, neurons project to the POA, the MnPO, the DMH, rMR, and IML to trigger vasodilation in cutaneous blood vessels. On the other hand, connections between the POA, MPO, thalamus, and somatosensory cortex are responsible for producing behavioral changes to promote heat dissipation and limit heat production. DH: dorsal horn of the spinal cord; DMH: dorsomedial hypothalamus; DRG: dorsal root ganglion; HPA: hypothalamic–pituitary–adrenal axis; IML: intermediolateral laminae; LPBd: dorsal part of the parabrachial nucleus; MPOa: medial preoptic area; MnPO: median preoptic area; POA: preoptic area; rMR: rostral medullary raphe; SAM: symptom–adrenomedullary axis; TRP: transient receptor potential.

**Figure 2 animals-14-01745-f002:**
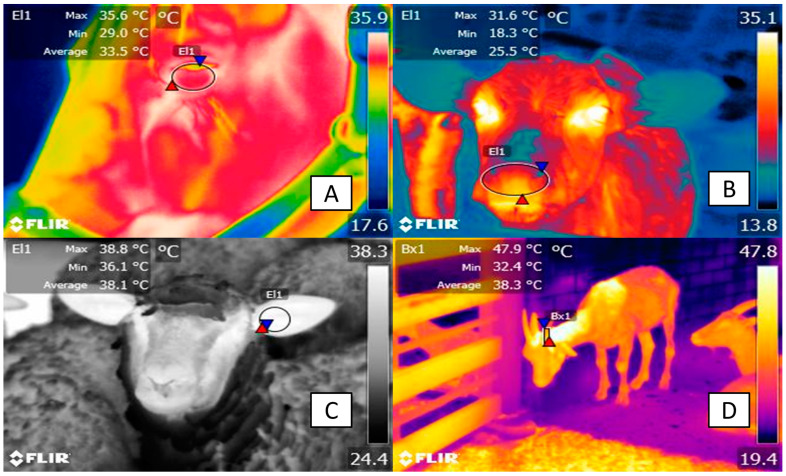
Recommend thermal windows in domestic animals for the recognition of thermal stress. (**A**) Ocular. This thermal window is recommended due to its irrigation of the infra and supraorbital artery, which helps detect changes in perfusion during states of hyperthermia. Likewise, this window has been shown to have a positive correlation with core body temperature. (**B**) Nasal. This region has been used to recognize changes in the respiratory pattern and frequency that are regularly modified during the perception of thermal stress. (**C**) Auricular. This window is irrigated by branches of the internal auricular artery (the marginal auricular and lateral auricular arteries). (**D**) Horn. This window has been recently proposed to evaluate the thermal state of animals due to the perfusion from the temporal artery that might represent a region for thermal exchange in livestock.

**Figure 3 animals-14-01745-f003:**
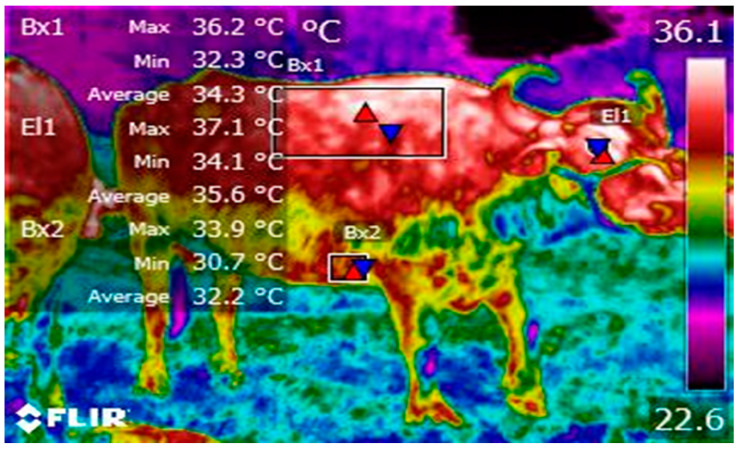
Thermal difference between superficial and central regions. The surface temperature of the dorsal (B × 1), axillary (B × 2), and ocular region (El1) is shown. In the dorsal region, a maximum temperature (red triangle) of 36.2 °C is observed, as well as an average of 34.3 °C and a minimum (blue triangle) of 32.3 °C. When comparing the average temperature of the dorsal region with the axillary region (B × 2), a difference of −2.1 °C was observed. In the orbital region (El1), the highest temperature can be observed, where the maximum, average, and minimum temperatures are above dorsal values by 0.9 °C, 1.3 °C, and 1.8 °C, respectively. This thermal difference between the regions may be due to the arrangement of the number of sweat glands and even the amount of blood capillaries present at a superficial level that affects the heat elimination capacity, which would facilitate the animals’ ability to recover thermoneutrality.

**Table 1 animals-14-01745-t001:** Physiological and behavioral changes in animals experiencing hyperthermia.

	Species Studied	Physiological Response and Behavioral Modifications	References
Vasodilation, sweating			
	Water Buffaloes	The number of glands per cm^2^ of skin is lower in buffalo (394 gland/cm^2^) than in cattle (2633 gland/cm^2^). This difference makes animals resort to polypnea as a heat-related tolerance response that also needs to be enhanced by providing shade.	[7]
	Water buffalo	After 2 h of transport, increases in the surface temperature above 4 °C were recorded with IRT for thermal windows: the head regions, such as the lacrimal caruncle, the periocular subregion, the lower eyelid, the ear canal, the nasal passages, and the parietal subregion, as well as body regions, such as the abdominal and lumbar regions.	[62]
	Review study	Peripheral vasodilation enhances heat dissipation and, therefore, helps to reduce core temperature.	[63,64]
	Review study	Sympathetic cholinergic nerves, vasoactive intestinal peptide, substance P, histamine, prostaglandins, and TRPV1 activation participate in vasodilation.	[65,66]
	Cow	The horns have a thermoregulatory function. In a temperate climate zone, the temperature in the horns increased by 0.18 °C for each unit of the heat load index.	[71]
	Cow	Arrangement of the blood vessels to thermoregulate is very important. For example, the parietal region is irrigated by the superficial temporal artery (*temporalis superficialis*), *transversa faciei*, *auricularis rostralis*, *palpebralis inferior lateralis*, and *palpebralis superior lateralis*, blood vessels that provide circulation to the horns, which can favor thermoregulation.	[72]
	Cattle	Stress-associated hyperthermia in lactating dairy cows during isolation challenges	[73]
	Sheep	hoof thermoregulation. A difference of 8.5 °C was determined between healthy and diseased hooves (diagnostic sensitivity of 92% and specificity of 91%).	[77]
	Sheep	The recording of thermal stress during shearing. Ocular surface IRT was positively correlated with rectal temperature and cortisol levels.	[78]
	Horses	Sweating dissipates 70% of the heat loss during exercise due to evaporation. Sweat rate (L/h/m^2^) in horses is three times greater in horses compared to humans with a similar exercise intensity.	[74,75]
	Water buffalo	The number of sweat glands/cm^2^ in the head, neck, and tail regions was 20–30% higher compared to the thorax and abdomen region. This difference might impact not only the effectiveness of thermoregulation strategies in animals but also make it evident that animals use specific anatomical regions to thermoregulate.	[90]
Endocrine and immunological response			
	Cattle	In hyperthermia due to acute heat stress, the concentrations of antidiuretic hormone, prolactin, glucocorticoids, and catecholamines increase, while those of aldosterone decrease. During chronic heat stress, levels of cortisol, growth hormone, and thyroxine may remain unchanged or may even decrease.	[60,94]
	Miniature pigs	Blood cortisol levels, which are associated with greater production of cytokines such as interleukin-4 (IL-4), IL-5, IL-6, and IL-12.	[99]
Thermoregulatory behaviors			
	Cows, goats, elephants	Behaviors that promote heat dissipation (e.g., shade-seeking, changes in body posture) or those that limit heat production (e.g., reducing feed intake and locomotor activity).	[84,85,86,87].
	Cows	Adaptative behaviors (increase time standing and reduce rumination time) from an average ambient temperature of 12 °C and a temperature–humidity index of 56.	[121]
	Ewes and Rams	Higher temperatures environments (35 °C) increased the total time of standing (355 vs. 486 min) and decreased eating (932 vs. 727 min). Likewise, the lying behavior of ewes and rams increased by up to 45.10 ± 2.42% during severe heat stress.	[123]
	Cows	An average rectal temperature of 39 °C, reduced dry matter intake (DMI) by 14.6 kg/day, and reduced and milk yields by 11.8 kg/day were recorded.	[106]
	Water Buffaloes	Wallowing helps to decrease body temperature by increasing heat dissipation via evaporation, as shown in a study assessing the thoracoabdominal surface temperature.	[135]

## Data Availability

Data sharing is not applicable.

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
