# Peer review of "Hypothalamic Neuromodulation and Control of the Dermal Surface Temperature of Livestock during Hyperthermia"

_animals, 2024, doi:10.3390/ani14121745_

Round 1

Reviewer 1 Report

Comments and Suggestions for Authors

The paper provides a comprehensive review of the hypothalamic control of hyperthermia in livestock, covering both physiological and behavioral responses. The authors have provided valuable insights into the mechanisms involved in thermoregulation. The following are some general weaknesses of the paper.

1. While comprehensive, the main body of the paper appears relatively concise for a review paper, containing less than 5,000 words.

2. The discussion of hyperthermia in livestock seems to be limited to ruminants, potentially overlooking other relevant species.

3. The absence of a table to summarize current research progress could hinder the reader's ability to grasp key findings efficiently.

4. The paper tends to describe previous studies without delving deeper into their implications or addressing potential methodological limitations, which could enhance the depth of analysis.

5. Although the paper briefly touches upon ongoing research concerning TRPM2 and TRPV1/TRPV3 channels, a more thorough exploration of their significance in understanding thermoregulation in livestock is warranted.

6. While focusing on physiological and behavioral responses, the paper lacks discussion on practical applications of this knowledge for managing heat stress in livestock production systems.

7. Given the paper's search methodology up to 2023, recent advancements in the field may not be adequately addressed. Incorporating any significant developments in animal science related to hyperthermia since 2023 would enhance the paper's relevance.

8. Improved clarity and organization, including clear subsection headings and a structured outline, would enhance readability and facilitate understanding of complex physiological pathways and mechanisms for readers.

Comments on the Quality of English Language

Minor editing of English language required.

Author Response

Response to Reviewer 1

The paper provides a comprehensive review of the hypothalamic control of hyperthermia in livestock, covering both physiological and behavioral responses. The authors have provided valuable insights into the mechanisms involved in thermoregulation. The following are some general weaknesses of the paper.

Response: The authors appreciate the time invested by the reviewer since our manuscript has improved significantly with his comments and observations. You can see the corrections in Yellow within our manuscript proposal.

  1. While comprehensive, the main body of the paper appears relatively concise for a review paper, containing less than 5,000 words.

Answer: Thank you for your observation. This review aims to discuss the hypothalamic control of hyperthermia in livestock, including the main physiological and behavioral changes that animals adopt to maintain their thermal stability. Therefore, the authors designed a concise strategy for the development of the topic. However, with the corrections and observations made by the reviewers, our manuscript has been strengthened in this aspect that you kindly pointed out to us.

  1. The discussion of hyperthermia in livestock seems to be limited to ruminants, potentially overlooking other relevant species.

Response: Thank you for your comment, in response to your suggestion, some paragraphs have been added where clarifications are made about the importance of local and systemic responses to hyperthermia, not only in cattle but also in sheep, horses, and rabbits. These changes are found in lines 213-220, 224-225, 228-230, 295-318, and 382-411.

  1. The absence of a table to summarize current research progress could hinder the reader's ability to grasp key findings efficiently.

Answer: Thank you for your comment, on line 496 a table has been added with the description of the main physiological and behavioral changes during hyperthermia.

  1. The paper tends to describe previous studies without delving deeper into their implications or addressing potential methodological limitations, which could enhance the depth of analysis.

Response: Thank you for your comment, the depth of the analysis has been improved with the inclusion of studies carried out on other species and the requested table.

  1. Although the paper briefly touches upon ongoing research concerning TRPM2 and TRPV1/TRPV3 channels, a more thorough exploration of their significance in understanding thermoregulation in livestock is warranted.

Response: Thank you for your comment, changes have been made according to your suggestion on lines 87-102, and 106-109.

  1. While focusing on physiological and behavioral responses, the paper lacks discussion on practical applications of this knowledge for managing heat stress in livestock production systems.

Response: Thank you for your comment, a section has been added regarding the topic you requested in lines 498-505.

  1. Given the paper's search methodology up to 2023, recent advancements in the field may not be adequately addressed. Incorporating any significant developments in animal science related to hyperthermia since 2023 would enhance the paper's relevance.

Response: thank you for your comment, some references from the 2023-2024 period have been added. Among them are the following:

  • Maia ASC, Moura GAB, Fonsêca VFC, Gebremedhin KG, Milan HM, Chiquitelli Neto M, Simão BR, Campanelli VPC, Pacheco RDL. Economically sustainable shade design for feedlot cattle. Front Vet Sci. 2023 Jan 25;10:1110671. doi: 10.3389/fvets.2023.1110671. 
  • Mincu M, Nicolae I, Gavojdian D. Infrared thermography as a non-invasive method for evaluating stress in lactating dairy cows during isolation challenges. Front Vet Sci. 2023 Sep 6;10:1236668. doi: 10.3389/fvets.2023.1236668
  • Dean L, Tarpoff AJ, Nickles K, Place S, Edwards-Callaway L. Heat Stress Mitigation Strategies in Feedyards: Use, Perceptions, and Experiences of Industry Stakeholders. Animals (Basel). 2023 Sep 26;13(19):3029. doi: 10.3390/ani13193029.
  • Silva WCD, Silva JARD, Martorano LG, Silva ÉBRD, Sousa CEL, Neves KAL, Araújo CV, Joaquim LA, Rodrigues TCGC, Belo TS, Camargo-Júnior RNC, Lourenço-Júnior JB. Thermographic Profiles in Livestock Systems under Full Sun and Shaded Pastures during an Extreme Climate Event in the Eastern Amazon, Brazil: El Niñoof 2023. Animals (Basel). 2024 Mar 11;14(6):855. doi: 10.3390/ani14060855.
  1. Improved clarity and organization, including clear subsection headings and a structured outline, would enhance readability and facilitate understanding of complex physiological pathways and mechanisms for readers.

Response: Thank you for your observations, the authors recognize that your comments have allowed us to improve the structure and organization of our manuscript proposal.

Reviewer 2 Report

Comments and Suggestions for Authors

Comments: 

• The review provides valuable insights into hyperthermia in livestock and the pivotal role of the hypothalamus in its regulation. It effectively synthesizes and discusses recent advancements in the field, albeit some information overlaps with existing literature.

• The paper is well-written and thoroughly elucidates its topics, complemented by elucidative figures.

• On page 2 (lines 86-91), there are inconsistencies regarding the activation of V1 and V2 in hot environments and V3 and V4 in warm environments. Specifically, the sentence on line 87 should replace the second occurrence of V1 with V2. Additionally, on line 88, TRPV2 is mentioned as important for temperatures above 34°C, which should likely be TRPV3 according to the authors cited in that paragraph.

• While most studies suggest animals increase their standing time as temperature rises to enhance heat dissipation, there is a discrepancy noted on page 9 (lines 363-366) where it states that lying time was higher in ewe, ram, and pig. If this observation is accurate, further clarification or potential explanations for these contradictory results are warranted.

I don't see the importance of inserting the word 'body posture' in the topic 5 (page 9). The word 'Behavioral changes' is enough as body posture change is part of behavioral changes.

Author Response

Response to Reviewer 2

  • The review provides valuable insights into hyperthermia in livestock and the pivotal role of the hypothalamus in its regulation. It effectively synthesizes and discusses recent advancements in the field, albeit some information overlaps with existing literature. 

Response: The authors thank the reviewer for all his comments, and we also thank him for the time he has invested in reviewing our manuscript proposal, since with his comments and observations, our paper has improved significantly. You can see the corrections in Blue within the manuscript.

  • The paper is well-written and thoroughly elucidates its topics, complemented by elucidative figures. 

Response: Thank you for your comment.

  • On page 2 (lines 86-91), there are inconsistencies regarding the activation of V1 and V2 in hot environments and V3 and V4 in warm environments. Specifically, the sentence on line 87 should replace the second occurrence of V1 with V2. Additionally, on line 88, TRPV2 is mentioned as important for temperatures above 34°C, which should likely be TRPV3 according to the authors cited in that paragraph. 

Response: Thank you for your comment, the data that caused the confusion, has been reviewed and we have corrected it. However, at the suggestion of reviewer 1, the paragraph has been complemented with information regarding TRPM2 and TRPV1/TRPV3 channels. These changes are found on lines 103-109.

  • While most studies suggest animals increase their standing time as temperature rises to enhance heat dissipation, there is a discrepancy noted on page 9 (lines 363-366) where it states that lying time was higher in ewes, rams, and pigs. If this observation is accurate, further clarification or potential explanations for these contradictory results are warranted.

Response: Thank you for your comment, an explanation of this contradictory effect has been added on lines 449-456.

I don't see the importance of inserting the word 'body posture' in the topic 5 (page 9). The word 'Behavioral changes' is enough as body posture change is part of behavioral changes.

Response: Thank you for your comment, the phrase on line 413 has been removed.

Round 2

Reviewer 1 Report

Comments and Suggestions for Authors

The paper is suitable for acceptance in its current form.